# Global Patch-wise Attention is Masterful Facilitator for Masked Image Modeling

## ABSTRACT

Masked image modeling (MIM), as a self-supervised learning paradigm in computer vision, has gained widespread attention among researchers. MIM operates by training the model to predict masked patches of the image. Given the sparse nature of image semantics, it is imperative to devise a masking strategy that steers the model towards reconstructing high-semantic regions. However, conventional mask strategies often miss these high-semantic regions or lack alignment with the masks and semantics. To solve this, we propose the Global Patch-wise Attention (GPA) framework, a transferable and efficient framework for MIM pre-training. We observe that the attention between patches can be the metric of identifying high-semantic regions, which can guide the model to learn more effective representations. Therefore, we firstly define the global patch-wise attention via vision transformer blocks. Then we design the soft-to-hard mask generation to guide the model gradually focusing on high semantic regions identified by GPA (GPA as a teacher). Finally, we design an extra task to predict GPA (GPA as a feature). Experiments conducted under various settings demonstrate that our proposed GPA framework enables MIM to learn better representations, which benefit the model across a wide range of downstream tasks. Furthermore, our GPA framework can be easily and effectively transferred to various MIM architectures.

## CCS CONCEPTS

• **Computing methodologies → Computer vision representations**.

## KEYWORDS

Self-supervised learning, Visual representation learning

## 1 INTRODUCTION

In recent years, self-supervised learning [4, 16] has received extensive attention in the field of computer vision. Its characteristic lies in the ability to learn meaningful representations without the need for any annotations. Inspired by the prominent self-supervised learning method in natural language processing known as Masked Language Modeling (MLM) [7, 34], Masked Image Modeling (MIM) [2, 12, 15] has also gradually gained prominence in self-supervised learning-based computer vision.

**Unpublished working draft. Not for distribution.**

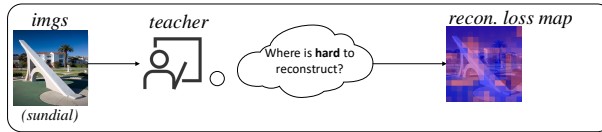

(a) Mask strategies based on reconstruction loss

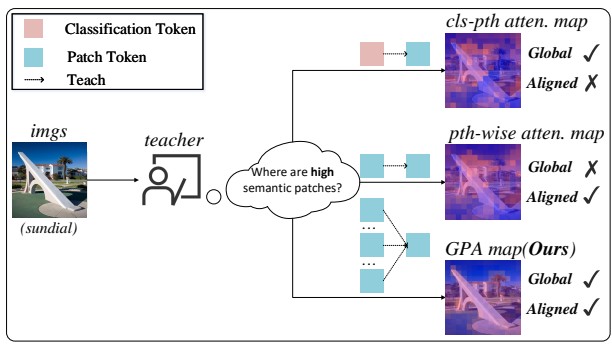

(b) Mask strategies based on attention

**Figure 1: Comparison between previous MIM mask strategies and our proposed GPA. (a) Mask strategies based on reconstruction loss. (b) Mask strategies based on attention. Top: attention between classification and patches. Middle: attention between single patch and other patches, *pth-wise atten.* indicates patch-wise attention. Bottom: Global Patch-wise Attention. Our proposed mask strategy, which is utilized in our framework. Firstly, we calculate attention for each patch pair and then aggregate them along the query dimension to derive the Global Patch-wise Attention.**

MIM operates by training the model to predict masked patches of the image. The masks act as the ***teacher***, while the model acts as the ***student***. The ***teacher*** guides the ***student*** to focus on the masked patches, in order to instruct and facilitate the ***student*** in understanding the entire image. Currently, the most common mask strategies are randomly select masked patches (random masking [15]), or randomly select a set of adjacent patches for masking (block-wise masking [2]). However, such mask strategies are evidently insufficient. Given the sparsity of semantic information in images, random masking tends to obscure patches with weak semantic content, thereby hindering the model's ability to learn valuable representations. Consequently, there is a need for improving the mask strategy, enabling the ***teacher*** to selectively teach the ***student*** to focus on high semantic patches. However, designing an effective mask strategy poses a significant challenge in MIM. Without supervised information to assist in distinguishing high semantic patches, the task of learning where to mask relies on the exploration of self-features within the image.

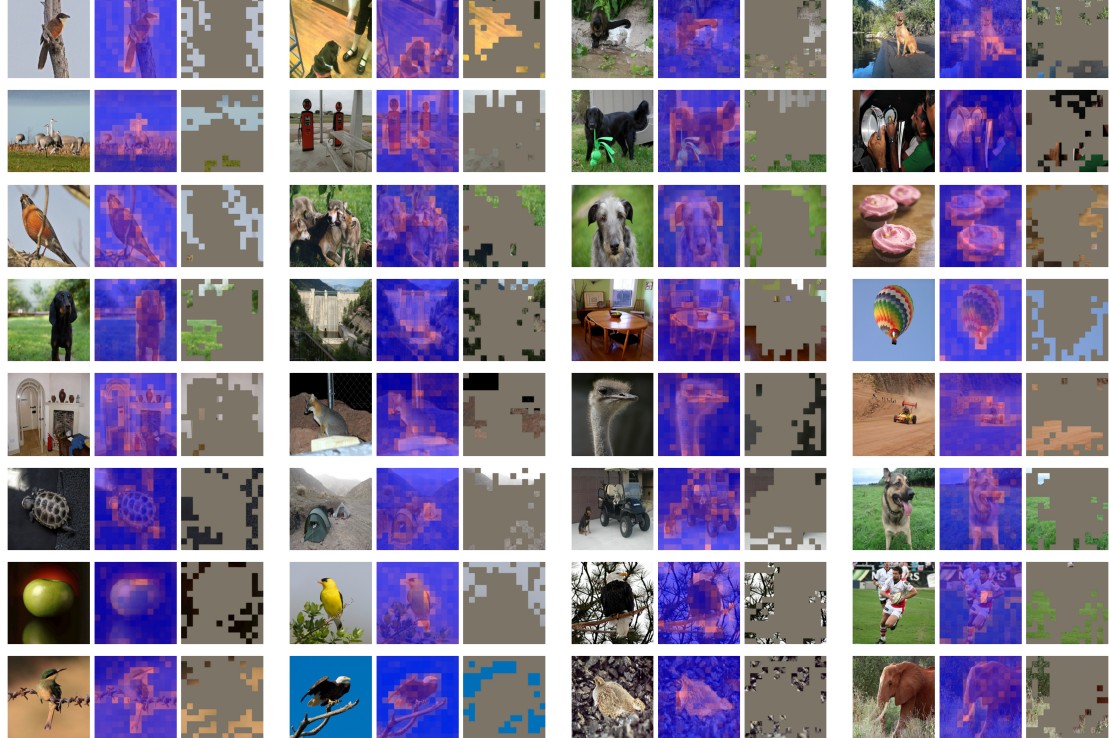

**Figure 2: Visualization of GPA on the ImageNet-1K validation set using the pre-trained ViT-B/16 model from MAE [15]. Each tuple displays the *input image* (left), *Global Patch-wise Attention* (center), and the *top 75% hard mask generated by GPA* (right), where red indicates high GPA.**

Figure 1 illustrates approaches of improving mask strategies, which can be divided into two main strategies: Figure 1(a) based on reconstruction loss [1, 36], where high semantic patches are believed to be difficult to reconstruct, so patches with large reconstruction loss are selected. The other strategy, shown at the top of Figure 1(b), is based on the attention between classification token and patch tokens (**cls-pth atten.**) of a pre-trained ViT [3, 19, 23, 28], as the classification token carries global semantics. By calculating the attention between classification token and each patch token, it is possible to locate which patch contains more semantic. However, for the reconstruction loss based mask strategies, we argue that there is not a strong correlation between high reconstruction loss and semantic. As shown in Figure 1(a), patches with high reconstruction loss are often at patches with rich RGB colors, which may often correspond to complex background colors in many images. As for the cls-pth attention based mask strategies, We argue that in the pre-training process of MIM, the cls-token does not directly participate in the reconstruction task, which leads to the semantic information of cls-tokens not being aligned with images. Therefore, using the cls-token directly might lead to semantic alignment issues with the mask. An alternative approach involves using CLIP, however, in this method, the cls-token's alignment with the text modality during pre-training could introduce modal alignment issues with the mask. Consequently, current attention-based masking strategies generally face alignment problems.

How can we find a self-feature that is **strongly associated with high semantic patches** and **aligned with the image** to guide the model in learning where to mask? In response to this challenge, we propose the **G**lobal **P**atch-wise **A**ttention pre-training framework (**GPA**) with a novel and natural mask strategy based on patch-wise attention as shown in the bottom of Figure 1(b). Inspired by attention's original definition, our strategy demonstrates a strong correlation with high semantic patches, aligning them with image features through attention computations between patches. Specifically, in the attention mechanism [35], the product of patch $A$'s *key* and patch $B$'s *query* quantifies how much patch $B$ attends to patch $A$. Consequently, the sum of products between the *key* of patch $A$ and the *query* of all other patches can be considered as the degree to which patch $A$ needs to be attended to in comprehending the entire image. We term this cumulative attention measure as **Global Patch-wise Attention (GPA)** for patch $A$. During the reconstruction process, patch tokens interact and gradually synchronize, aligning their learned features. This alignment allows the Global Patch-wise Attention (GPA) we formulated to target patches with significant semantic content accurately. According to our framework, patches that exhibit higher GPA are given priority for masking, highlighting their crucial semantic importance.

Consistently masking patches with high GPA could hinder the model's ability to learn environmental context. To address this, we introduce a soft-to-hard mask generation approach. Contrasting

with the easy-to-hard strategy [36] that progressively presents more difficult challenges to the model, our method transitions from a soft ordering (sampling) to a hard ordering (sorting) of masks, thereby progressively directing the model's focus towards critical entities.

Previous studies have identified various features, like HOG [39] and reconstruction loss [36], as effective targets for reconstruction. The sensitivity of our proposed GPA to semantics suggests that employing GPA as a reconstruction feature could offer a direct signal for the model to discern the semantic density of patches. Consequently, we introduce a new task for MIM, termed Masked Attention Modeling (MAM), that utilizes GPA for reconstruction. Empirical evidence indicates that MAM facilitates the model in acquiring more effective representations.

Our contributions are following:

- We propose a **patch-wise attention-based** mask strategy that can indicate high semantic patches to mask and can easily transfer to any Transformer-based MIM.
- We design the **MAM task** to explore the potential of employing GPA as a feature, demonstrating its effectiveness in enhancing the model's performance across various metrics.
- We conduct extensive experiments across multiple benchmarks to confirm the **effectiveness** of our GPA framework, and our transfer experiments on various MIM frameworks demonstrate the **universality** of our masking strategy.

## 2 RELATED WORK

**Self-supervised learning.** Self-supervised learning, eliminating the need for annotated data, fosters the development of universal and meaningful representations by harnessing supervisory information inherent in the data itself. This paradigm has achieved remarkable success across various domains [7, 31, 34]. In the field of Computer Vision, self-supervised learning [8, 13, 22, 33, 47] primarily encompasses two main directions: Masked Image Modeling [2, 15] and Contrastive Learning [14, 16, 38, 45]. Masked Image Modeling involves training a model to predict the original signals in intentionally obscured regions of an image, thereby aiding in learning rich representations by reconstructing these masked patches. Contrastive learning employs instance discrimination [41] tasks to bring positive samples closer in the feature space while pushing negative samples farther apart [4].

**Masked image modeling.** Inspired by the success of Masked Language Modeling [7], Masked Image Modeling, The mirrored approach of MLM in computer vision, has received widespread attention [9, 21, 32, 39, 43, 46]. There are currently two paradigms: continuous [15] and discrete [2]. The continuous paradigm maps images to a continuous embedding space, while the discrete paradigm questions the suitability of continuous space for image reconstruction. Therefore, these methods map patch tokens to a discrete embedding space (codebook) using similarity calculations. Both paradigms train a Vision Transformer [10, 30] to predict predefined image features (such as RGB [15], HOG [39]) from their respective spaces.

**Mask strategies.** Language exhibits a remarkably high level of semantic density, which means that in masked language modeling, random masking is highly likely to cover tokens with high semantic significance [7]. In contrast, the semantic distribution in images is considerably sparse and often concentrated on specific entities. Therefore, the design of mask strategies becomes crucial in the context of images. Currently, mask strategies can be broadly categorized into three main types. **(1) Random** [2, 15, 24]. MAE [15] initiates MIM with a high mask ratio by randomly masking image patches, while BEIT [2] employs block-wise masking inspired by n-gram masking in MLM. UM-MAE [24] masks one patch in each 2×2 local window, enabling pyramid-based ViTs (e.g., PVT [37], CoaT [44], and Swin [29, 30]) to take the random sequence of partial vision tokens as input. **(2) Reconstruct Loss**. AdaMAE [1] operates under the assumption that high-semantic patches are challenging to reconstruct, thus masking patches with high reconstruction loss to facilitate model learning of these semantically rich areas. HPM [36] aims to provide the model with challenging tasks, specifically targeting the patches that are difficult for the model to reconstruct and design a "easy-to-hard" mask generation. **(3) Class-to-patch attention**. These methods consider the class token to carry summarizing semantics. SemMAE [23], AutoMAE [3], AttMask [19] and AMT [28] utilize the attention between the class token and different patch tokens. SemMAE [23] leverages this attention mechanism to pinpoint specific semantic patches, progressively masking these areas from partial to complete coverage, while also training an additional StyleGAN [20]-based decoder distilled by iBOT [49]. AutoMAE and AttMask generate an attention map based on this similarity to derive the mask strategy. MILAN [18] employs knowledge distillation by leveraging the fact that the class token of the CLIP image encoder, which is trained with a large amount of textual modality data during pre-training, contains global semantic information. It identifies high-semantic patches and applies masking by calculating the similarity between the CLIP image encoder and the patch encoder.

## 3 METHOD

In this section, we initially explore preliminaries and the background on vision transformers in Section 3.1. Subsequently, we delve into the computation of **Global Patch-wise Attention (GPA)** in Section 3.2, discussing its application as a teacher to guide mask strategy formulation for enhancing model learning. In Section 3.3, we examine how GPA serves as a feature, assisting the model in its learning process. Lastly, in Section 3.4, we investigate the impact of different heads on masking through an attention selection module. Figure 3 presents an overview of our proposed GPA framework.

### 3.1 Preliminaries

*3.1.1 Masked Autoencoder.* Given an input image $X \in \mathbb{R}^{H \times W \times C}$, where $H$ and $W$ are the height and width and $C$ is the number of channels. The first step is patchify, where the image is divided into $n$ patches, each of size $P$, where $n = HW/P^2$ and the patches $X_{pi} \in \mathbb{R}^{P \times P \times C}$, $i = 1, 2, ..., n$. Each patch is flattened and projected to an embedding vector $z_i \in \mathbb{R}^D$, following the practice of transformer-based model, a learnable embedding $z_{[cls]} \in \mathbb{R}^D$ representing the classification token is prepended to the beginning of the entire vector:

$$Z = (z^{[cls]}; z^1; ...; z^n) \in \mathbb{R}^{(n+1)D}. \quad (1)$$

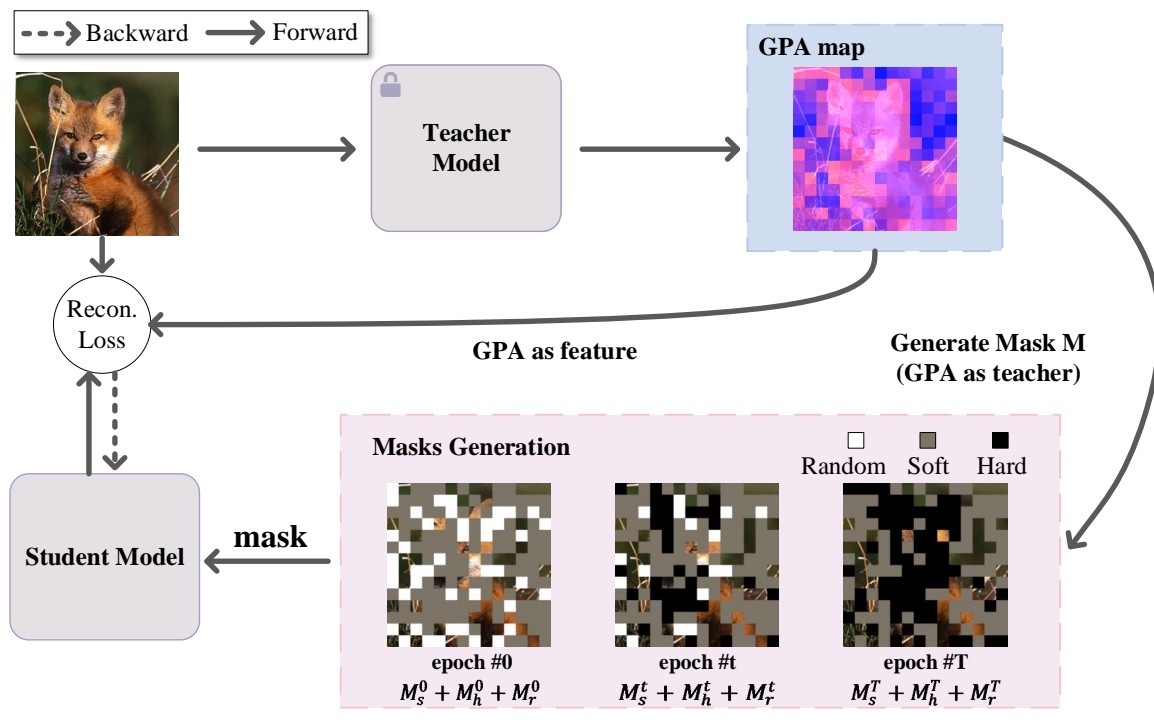

**Figure 3: Illustration of our proposed GPA pre-training framework. During pre-training, each image is firstly fed into teacher model to generate a soft-to-hard binary mask on current epoch. Subsequently, the visible patches are input into the student model, which reconstructs the RGB feature and the Global Patch-wise Attention feature.**

Subsequently, these patch embeddings are either retained or masked according to a mask vector $M = 0, 1^n$, where 0 signifies masking and 1 signifies retention, and the embedding of the image after patchify and masked goes to:

$$\widetilde{Z} = M \odot Z + (1 - M) \odot Z^{[mask]}, \quad (2)$$

where $Z^{[mask]}$ is a learnable embedding [15] of all masked token. Ultimately, the masked autoencoder is trained to extract meaningful representations by reconstructing these masked patches:

$$L_{RGB} = \sum_{i=1}^{n} (1 - M_i) * \mathcal{M}(\mathcal{F}(\widetilde{Z}_i), Z_i), \quad (3)$$

where $\mathcal{F}(\cdot)$ and $\mathcal{M}(\cdot, \cdot)$ represent model and its loss function, $(1 - M)$ represents that only the masked tokens require loss computation, indicating that the model learns representations by reconstructing the masked patches.

Extensive previous experiments [1, 36] support our assertion that mask strategy plays a pivotal role in representation learning. As previously mentioned, we aim to identify a mask strategy that possesses a strong semantic correlation and alignment with the image.

*3.1.2 Multi-Head Self-Attention (MSA)..* Given an input embedding $Y \in \mathbb{R}^{(n+1)D}$, a MSA [35] layer uses three linear layers to project $Y$ to its query $Q_h$, key $K_h$ and value $V_h$ embedding, where $h = 1, ..., H$ and H is the number of heads, $Q_h, K_h, V_h \in \mathbb{R}^{(n+1)\frac{D}{H}}$, in a MSA

layer, the attention received by the i-th patch from the j-th patch is denoted as:

$$Att_h^{ij} = \frac{Q_h^i K_h^j}{\sqrt{D/H}}. \quad (4)$$

## 3.2 Global Patch-wise Attention as a Teacher

Ideally, one might assume that semantic information is randomly distributed across an image, making a random mask strategy seem effective. Yet, in practice, the distribution of semantic content is far from random. For downstream tasks in image understanding, such as image classification and semantic segmentation, the required semantics are typically concentrated on specific entities, and a random mask strategy fails to capture this distribution effectively. To address the disparity in semantic perception between pre-training on a large amount of unlabeled data and finetuning on downstream tasks, we propose a "masked by global patch-wise attention" strategy. Our idea is that these semantically significant entities are the areas we naturally focus on when viewing an image. Hence, by incorporating attention mechanisms into the mask strategy, we can naturally capture the distribution of image semantics.

*3.2.1 Global Patch-wise Attention.* To begin, we define the **Global Patch-wise Attention** that the m-th patch receives within the

entire image in a MSA layer:

$$GPA_h^m = \sum_{i=1, i \neq m}^{n} Att_h^{mi}, \qquad (5)$$

where $GPA_h^m$ represents the Global Patch-wise Attention received by m-th patch in h-th MSA layer. The sum of patch-level attention over the entire image, denoted as $\sum^i Att^{mi}$, implies the importance of attending to the m-th patch when comprehending the image.

Utilizing GPA, we can target high semantic patches for masking by applying the $argsort(\cdot)$ operation on $GPA_h$ in descending order, pinpointing patches with rich semantics, as shown in Figure 2, the GPA map can precisely indicate high-semantic patches. As a teacher, masking should not entirely obscure high-semantic patches initially but should progressively reveal what the model needs to learn. This approach mirrors the process of human learning, which often starts with simpler tasks before progressing to more complex ones. Masking entities from the start prevents the model from learning the associated information across different semantic patches, which is equally crucial for accurate downstream tasks.

*3.2.2 Soft-to-hard Mask Generation.* Given the potential issue of masking high semantic patches prematurely, which could hinder the model's learning, how can we develop a mask generation strategy that aligns with the gradual learning curve? To address this, we propose a Soft-to-Hard mask generation approach. The *argsort* operation masks high semantic patches absolutely, we regard it as a "hard" strategy, to employ a "soft" strategy, we sample each patch with a probability distribution defined by GPA. As the training epochs increase, we gradually increase the proportion of hard masks while decreasing the proportion of soft masks.

The number of soft and hard mask patches is denoted as:

$$\alpha_t^s = \alpha_0^s + \frac{t}{T} \cdot (\alpha_T^s - \alpha_0^s),$$
$$\alpha_t^h = \alpha_0^h + \frac{t}{T} \cdot (\alpha_T^h - \alpha_0^h), \qquad (6)$$
$$\alpha_t^r = 1 - (\alpha_t^s + \alpha_t^h)$$

For each epoch $t = 1, ..., T$, the training progress is $t/T$. Here, $\alpha_0^s$ and $\alpha_T^s$ denote the initial and final soft mask ratios, respectively, which decrease gradually from $\alpha_0^s$ to $\alpha_T^s$. Conversely, the hard mask ratio incrementally rises from $\alpha_0^h$ to $\alpha_T^h$. This dynamic adjustment from soft to hard masks enables the model to progressively focus on learning from more semantically significant regions.

## 3.3 Global Patch-wise Attention as a Feature

To leverage GPA as a feature, we introduce the MAM task for the decoder, which aims to predict GPA. This task employs the Mean Squared Error (MSE) as its loss function. Here, the decoder strives to align its attention predictions with the GPA, guiding the model to better grasp the underlying semantics of the image.

$$L_{GPA} = \sum_{i=1}^{n} (1 - M_i) * (\mathcal{F}'(\widetilde{Z}_i) - GPA_i)^2, \qquad (7)$$

where $\mathcal{F}'(\cdot)$ represents encoder and attention decoder, GPA is a ground-truth for attention prediction. The RGB and attention decoder work in an alternating way, and encourage the encoder to learn better representations.

$$Loss = L_{RGB} + L_{GPA} \qquad (8)$$

## 3.4 Attention Selection

It is evident that randomly choosing a single head may not encompass global semantic. To solve this problem, we design a **Few-shot Attention Selection** module to select the head that encapsulates global semantic, as shown in Figure 4.

In most computer vision tasks, it is essential for the model to attend to the specific patches in visual information that contain entities, which aligns closely with our concept of masked by GPA. Hence, we can enhance the masking of high-information patches by providing the model with few downstream task samples before pre-training. This augmentation facilitates the mask by attention approach in effectively capturing the high-information patches, which can improve our mask strategy. To achieve this, We randomly select a subset $(X_1^R, y_1^R; ...; X_{n'}^R, y_{n'}^R)$ of $n'$ samples from the ImageNet classification task, which $X_R$ serves as a subset of the pre-training data $X$ and $y_R$ represents the labels for classification.

Firstly, for each $X_i^R$, We manually annotate the corresponding patches in the image based on its label $y_i^R$. We consider the annotated patches of all images $M_{gt}$ as the ground truth for the mask strategy. Then For each head in each ViT block, we generate its mask strategy $M_h$ by $argsort(GPA)$. Finally, we evaluate the mask strategy of each head using $m = Recall(M_{gt}, M_h)$ and utilize $argmax(m)$ to select the best-performing head. This selected head serves as the guiding teacher for the model's learning process.

## 4 EXPERIMENTS

**Setups.** In the experiment setups, all tests are executed on 2 × NVIDIA GeForce RTX 3090s, using the Vision Transformer as the backbone and pre-training on the ImageNet-1K dataset for 200 epochs. The optimization is done using AdamW with an initial learning rate of 1.5e-4 and a batch size of 4096. Input images are resized to 224×224 and segmented into 16×16 patches, following the methodology of MAE. Our implementation is based on MAE [15].

**ImageNet classification.** For ImageNet classification, we assess our GPA through end-to-end fine-tuning across 100 epochs, utilizing AdamW for optimization with a learning rate of 5e-4, a batch size of 1024, a layer decay of 0.8, and a cosine schedule for learning rate decay, aiming to improve Top-1 accuracy on the validation set. We report **Top-1** accuracy on the validation set. **COCO object detection and instance segmentation.** In COCO object detection and instance segmentation [27], we apply Mask R-CNN [17] with FPN [26] on the COCO dataset, conducting end-to-end fine-tuning over a 1× schedule (12 epochs) at a 1024×1024 resolution. The performance is measured using $AP_{box}$ and $AP_{mask}$ metrics, based on the ViTDet [25] framework and detectron2 [40].

**ADE20k semantic segmentation.** We experiment on ADE20K [48] using UperNet [42], and perform end-to-end fine-tuning with 80k iterations for ablations and 160k iterations for comparisons. The resolution is 512 × 512. We take mIoU [11] as the evaluation metric. Our implementation is based on mmsegmentation [6].

## 4.1 Ablation Study

We study different mask strategies and structures in this section, by default, the overall masking ratio is $\alpha = 0.75$, for mask generation, $\alpha_0^s = 0.5$, $\alpha_T^s = 0.0$ for soft masking, and $\alpha_0^h = 0.0$, $\alpha_T^h = 0.5$ for hard masking, the definition of different $\alpha$ can be found in Equation 6, and for the default structure in mask ablations, we only use RGB features, this allows us to eliminate the influence of GPA as a feature on the model, thereby directly verifying the effectiveness of GPA as a teacher. Default settings of our framework are marked in grey .

**Table 1: Ablation study on different roles of GPA. We study GPA as a teacher and as a feature. All cases are pre-trained 200 epochs on ImageNet with ViT-B.**

| Model | as teacher | as feature | finetune acc (%) |
|-------|-----------|-----------|------------------|
| MAE | - | - | 82.23 |
| GPA | ✓ | - | 82.73 (**+0.50**) |
|  | ✓ | ✓ | 83.22 (**+0.99**) |

*4.1.1 Global Patch-wise Attention in various roles.* To substantiate the capabilities of GPA in both the teacher and feature, we conducted a study involving different roles. When GPA employed as a teacher, we ranked GPA for each patch and utilized parameters from the default setting to generate soft-to-hard masks. When GPA utilized as a feature, we augmented the model to incorporate GPA as a reconstructive feature.

As illustrated in Table 1, our GPA framework significantly enhances the model's performance. Specifically, when GPA serves both as a teacher and a feature, the fine-tuning Top-1 accuracy reaches 83.22%, outperforming the baseline by +0.99%. When GPA is solely employed as a teacher, the accuracy is 82.73%, which is a 0.50% improvement over MAE, which proves that **GPA is masterful facilitator for masked autoencoders**.

**Table 2: Ablation study on downstream tasks. We study GPA on different datasets and tasks. All cases are pre-trained 200 epochs on ImageNet with ViT-B.**

| Model | as teac | as feat | COCO $AP_{box}$ | $AP_{mask}$ | ADE20k mIoU |
|-------|---------|---------|-----------------|-------------|-------------|
| MAE | - | - | 33.0 | 29.8 | 40.5 |
| GPA | ✓ | - | 33.3(**+0.3**) | 30.1(**+0.3**) | 41.6(**+1.1**) |
|  | ✓ | ✓ | 34.5(**+1.5**) | 31.1(**+1.3**) | 44.3(**+3.8**) |

In downstream tasks, as shown in Table 2, the GPA-enhanced model exhibits significant improvements: +1.5 in $AP_{box}$ and +1.3 in $AP_{mask}$ on COCO, and +3.8 in mIoU on ADE20k compared to the baseline. This improvement is notable even when GPA is only used as a teacher, demonstrating its robust transferability across various tasks and datasets.

Notably, only taking GPA as teacher outperforms MAE by +0.3 $AP_{box}$ and +0.3 $AP_{mask}$ on COCO, and +1.1 mIoU on ADE20k, which demonstrates the strong transfer ability of GPA across different datasets.

*4.1.2 Soft-to-hard Masking.* To prove soft-to-hard mask generation does bring better performance, we study various masking generations in Table 3 and Table 4. We gradually adjusted the "hardness" ("softness") of the mask by tuning parameters $\alpha_0^h$ and $\alpha_T^h$ ($\alpha_0^s$ and $\alpha_T^s$). For instance, in the experiments represented by $\alpha_0^s = 0.5$ and $\alpha_T^s = 0$ in Table 3, masks are "softer" compared to experiments with $\alpha_0^s = 1$ and $\alpha_T^s = 1$. We observe that a harder mask generation prompt the model to focus solely on a small region, while a softer mask generation approaches random masking.

**Table 3: Ablation study on soft masking. We study the effect of different $\alpha_0^s$ and $\alpha_T^s$.**

| hardness | $\alpha$ | $\alpha_0^s$ | $\alpha_T^s$ | finetune(%) |
|----------|----------|--------------|--------------|-------------|
| random | 0.75 | 0 | 0 | 82.69 |
| ↓ | 0.75 | 0.5 | 0 | 82.73 (**+0.04**) |
| ↓ | 0.75 | 1 | 0 | 82.69 (-) |
| soft | 0.75 | 1 | 1 | 82.70 (**+0.01**) |
| random | 0.5 | 0 | 0 | 82.50 |
| ↓ | 0.5 | 0.5 | 0 | 82.54 (**+0.04**) |
| soft | 0.5 | 1 | 1 | 82.53 (**+0.03**) |
| random | 0.9 | 0 | 0 | 82.41 |
| ↓ | 0.9 | 0.5 | 0 | 82.51 (**+0.10**) |
| soft | 0.9 | 1 | 1 | 82.37 (**-0.04**) |

Table 3 demonstrates the impact of soft masks on the model. Overall, soft masks offer a modest gain for the model, with improvements in top-1 fine-tuning accuracy not exceeding 0.1%. Among these, the group with $\alpha_0^s = 0.5$ and $\alpha_T^s = 0$ consistently performs the best across different mask ratios.

**Table 4: Ablation study on hard masking. We study the effect of different $\alpha_0^h$ and $\alpha_T^h$.**

| hardness | $\alpha$ | $\alpha_0^h$ | $\alpha_T^h$ | finetune(%) |
|----------|----------|--------------|--------------|-------------|
| soft | 0.75 | 0 | 0 | 82.25 |
| ↓ | 0.75 | 0 | 0.5 | 82.73 (**+0.48**) |
| ↓ | 0.75 | 0 | 1 | 82.71 (**+0.46**) |
| hard | 0.75 | 1 | 1 | 82.27 (**+0.02**) |
| soft | 0.5 | 0 | 0 | 82.22 |
| ↓ | 0.5 | 0 | 0.5 | 82.54 (**+0.32**) |
| hard | 0.5 | 0 | 1 | 82.37 (**+0.15**) |
| soft | 0.9 | 0 | 0 | 82.20 |
| ↓ | 0.9 | 0 | 0.5 | 82.51 (**+0.31**) |
| hard | 0.9 | 0 | 1 | 81.82 (**-0.38**) |

Table 4 presents the influence of hard masks on the model. Compared with soft masks, hard masks play a pivotal role in model performance. Inferior hard mask strategies can even result in detrimental effects on the model's learning. Across all mask ratios, the group with $\alpha_0^h = 0$ and $\alpha_T^h = 0.5$ consistently outperforms others.

It is noteworthy that when $\alpha_0^h = 1$ and $\alpha_T^h = 1$, all masks are hard masks. Under this setting, if the mask ratio $\alpha$ is sufficiently large, the

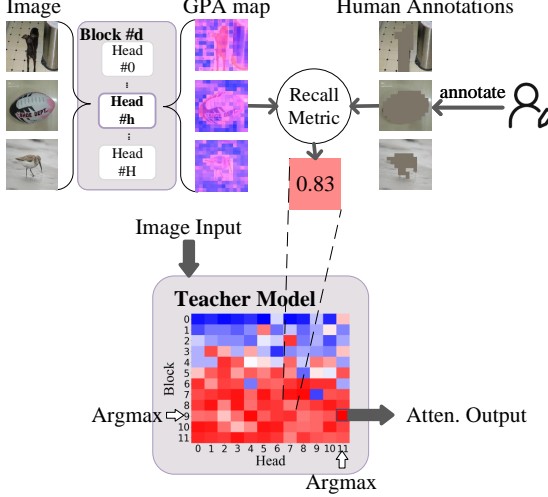

Figure 4: GPA recall on different blocks.

model is virtually unable to perceive any entities, necessitating that it learns the image semantics from an abundance of meaningless environmental information. Consequently, as $\alpha$ increases to 0.9, an excessive hard mask can paradoxically have a detrimental impact on the model's performance.

In conclusion, we find that enhancing the hardness of the masking approach does not uniformly result in improved performance. Maintaining an element of randomness in the masking process generally produces more favorable results. Increasing mask rigidity suggests that the model predominantly perceives background elements, and as illustrated in Figure 2, this can challenge even humans in reconstructing masked entities solely from background details.

Table 5: Ablation study on attention selection. We study the effect of different method, *first/last* represents the first/last head of a ViT, b0h0 means the #0 head of #0 block in Figure 4, -1 means the sum of all heads.

| method | first | last | all | 1 shot | 10 shots | 50 shots |
|---|---|---|---|---|---|---|
| head | **b**0**h**0 | **b**11**h**11 | **b**11**h**-1 | **b**10**h**10 | **b**9**h**11 | **b**9**h**11 |
| acc(%) | 82.31 | **82.70** | 82.34 | 82.68 | **82.73** | 82.73 |

*4.1.3 Attention Selection.* In addition, for the few-shot attention selection module, we explored various methods for choosing teacher heads in Table 5.

It is evident that optimal head selection (**b**9**h**11) can be achieved with just a **small number of images** (10 shots). Furthermore, a high finetuning accuracy (82.70%) can be attained without the need for any shots by selecting only the last head (**b**11**h**11).

## 4.2 Comparison with Previous Methods

**Image Classification.** We compare our proposed GPA with a wide range of mask strategies using top-1 fine-tuning accuracy on ImageNet-1K in Table 6 and linear probing in Table 7. Where selected methods can be summarized into three streams: (1) masked by random [2, 15, 24]. (2) masked by reconstruct loss [36]. (3) masked by cls-pth attention [3, 18, 23, 28]. All methods are evaluated under the same backbone (ViT-B/16).

Table 6: Comparison with state-of-the-art mask strategies on ImageNet-1K, all methods are evaluated by finetuning. The best results are shown in boldface.

| method | epochs | finetune(%) |
|---|---|---|
| *masked by random* | | |
| MAE [15] | 200 | 82.2 |
| UM-MAE [24] | 200 | 82.8 |
| BEiT [2] | 800 | 83.2 |
| *masked by recon. loss* | | |
| HPM [36] | 200 | 83.0 |
| *masked by cls-pth atten.* | | |
| AMT [28] | 400 | 82.8 |
| SemMAE [23] | 800 | 83.3 |
| AutoMAE [3] | 800 | 83.3 |
| MILAN[18] | 400 | 83.3 |
| *masked by patch-wise atten.* | | |
| GPA [**Ours**] | 200 | 83.2 |
| **GPA [Ours]** | **300** | **83.4** |

It is notable that our model achieved Top-1 accuracies of 83.2% and 83.4% on ImageNet after just 200 and 300 epochs of pre-training, respectively. This performance surpasses that of competing models that require more pre-training epochs, thereby underscoring the efficiency and effectiveness of our GPA framework.

Table 7: Comparision with SOTAs on linear probing.

| | MAE [15] | HPM [36] | CAE [5] | GPA |
|---|---|---|---|---|
| Acc(%) | 50.8 | 54.9 | 64.1 | **64.9** |

In linear probing tasks, our method utilizes the representations derived from self-supervised learning without tweaking any pre-trained parameters. The outcomes, as showcased in Table 7, reveal that our framework's representations not only achieve optimal effectiveness but also outperform the SOTA masking strategies by a significant margin of up to 10.0%. This underscores the GPA framework's robust capability in modeling visual features.

**Semantic Segmentation**. We experiment our GPA framework on ADE20k using UperNet for 160k iterations in Table 8. Our model is pre-trained 800 epochs on ImageNet-1k with ViT-B/16, consistent with other models. From the table, the GPA framework significantly improves performance over supervised pre-training by +1.2 mIoU. More importantly, GPA outperforms self-supervised methods under all settings.

**Efficiency of GPA.** From an efficiency perspective, GPA demonstrates an exemplary balance between computational demand and

**Table 8: Comparison with state-of-the-art mask strategies on ADE20k semantic segmentation using UperNet, The best results are shown in boldface.**

| method | | mIoU |
|---|---|---|
| MoCo v3 | [ICCV'21] | 47.3 |
| BEiT | [ICLR'22] | 47.1 |
| MAE | [CVPR'22] | 48.1 |
| SemMAE | [NIPS'22] | 46.3 |
| HPM | [CVPR'23] | 48.5 |
| **GPA** | [Ours] | **48.9** |

performance enhancement. With a FLOPs ratio of 1.1×, GPA exhibits a marginal computational overhead compared to MAE (1.0×), and notably, it maintains a considerable efficiency advantage over HPM (1.5×). This indicates that GPA's superior performance is not at the expense of significant computational resource increment, showcasing its proficiency in leveraging computational resources effectively. Furthermore, the learning parameter footprint of GPA

**Table 9: Comparison of Efficiency on different framework.**

| method | fine-tune(%) | lin. prob.(%) | FLOPs | learn. para. |
|---|---|---|---|---|
| MAE [15] | 82.2 | 50.8 | 1.0× | 1.0× |
| HPM [36] | 83.0 | 54.9 | 1.3× | 1.2× |
| GPA | 83.2 | 64.9 | **1.1×** | **1.0×** |

is aligned with MAE at 1.0×, underscoring GPA's ability to optimize performance without escalating the model complexity. In contrast, HPM's learning parameter size stands at 1.2×. Thus, GPA underscores its architectural efficiency, optimizing performance within the confines of constrained computational and parameter resources, which is a crucial consideration in model scalability and deployment.

**Qualitative results.** We provide qualitative results on COCO [27] validation set in Figure 5, we pre-train the model for 200 epochs on ImageNet-1k, which means the model has never seen this datasets. We found that the model is still able to effectively identify high-semantic regions that contribute to downstream tasks.

## 4.3 Applying our mask strategy to other architectures

To further elucidate the adaptability and the performance-enhancing capability of our mask strategy across various MIM architectures, we integrated our masking approach, which does not introduce additional learnable parameters, into different models. Remarkably, as delineated in Table 10, our mask strategy, when applied to MAE-B and ConvMAE-S, both of which were pre-trained for 200 epochs and fine-tuned for 100 epochs, yielded significant improvements. Specifically, the application of the GPA mask strategy on MAE-B resulted in an increment of 0.5% in the ImageNet Top-1 accuracy, demonstrating a subtle yet positive enhancement. More notably, the implementation on ConvMAE-S led to a substantial enhancement of 2.4% in the same metric.

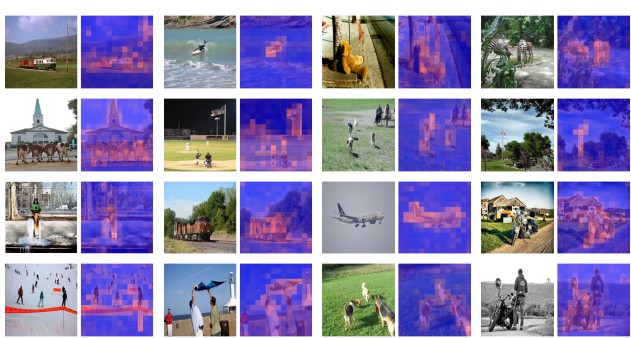

**Figure 5: Visualization on COCO validation set. For each tuple, we show the image (left) and GPA (right).**

Furthermore, when extending our mask strategy to other models like CAE-S and iBOT-B, which were pre-trained for a shorter duration of 50/100 epochs, we observed consistent improvements in performance. Both models exhibited an increase of 1.1% in K-NN accuracy post the integration of our masking approach.

**Table 10: Comparison of applying GPA as a teacher to other models before and after. We apply our mask strategy on MAE, ConvMAE, CAE and iBOT. ★: Evaluate on K-NN accuarcy.**

| method | epochs | Baseline | GPA as teacher |
|---|---|---|---|
| MAE-B [15] | 200 | 82.2% | 82.7% (**+0.5%**) |
| ConvMAE-S [12] | 200 | 80.0% | 82.4% (**+2.4%**) |
| CAE-S★ [5] | 50 | 37.4% | 38.5% (**+1.1%**) |
| iBOT-B★ [49] | 100 | 47.3% | 48.4% (**+1.1%**) |

These findings underscore the generalizability and effectiveness of our mask strategy in boosting the performance of various MIM architectures, suggesting its potential as a versatile and potent tool in enhancing model accuracy without complicating the learning parameter landscape.

## 5 CONCLUSION AND DISCUSSION

In this paper, we aim at identifying an feature for recognizing high-semantic regions and improving mask strategy in masked image modeling. To this end, we design a highly intuitive and simple method for computing semantic density and utilize it to propose a novel GPA framework, which is global and aligned. The mask strategy in GPA framework can be applied to any ViT-based masked image modeling approach and consistently improves performance. We validated the effectiveness of our method on different architectures and downstream tasks.

Although the GPA framework does not introduce additional learnable parameters, it integrates a teacher ViT, which leads to increased memory and computational requirements. Future research could focus on devising a framework that achieves similar objectives without the necessity for an additional ViT.

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
