# OpenReview forum: "Global Patch-wise Attention is Masterful Facilitator for Masked Image Modeling"
_acmmm.org/ACMMM/2024/Conference — MM2024 Poster_

### Official Review · Reviewer_8Cws · 2024-05-24

**Rating:** 4
**Confidence:** 3

**Summary:**

The paper presents a novel framework called Global Patch-wise Attention (GPA) for masked image modeling (MIM) in computer vision. GPA aims to enhance the model's ability to predict masked patches by focusing on high-semantic regions. It leverages attention between image patches to identify these regions, guiding the model to learn more effective representations. The GPA framework includes a soft-to-hard mask generation strategy and an additional task for predicting GPA.

**Strengths:**

- This paper is clear and well-written.
- The visualization results show that GPA can distinguish the foreground and the subject very well.
- Soft-to-hard Masking sounds reasonable to me, similar to curriculum learning.
- The experiments indicate that this method provides improvements in MIM approaches across various transformer architectures.

**Limitations:**

- The paper does not seem to be very closely related to multimodality or multimedia. And given that the method can be easily adapted to any masking method based on transformer architecture, could GPA be applied to other fields, such as video?
- It would be better to include some knowledge distillation papers in related work.
- Since a softer mask generation approaches random masking, the use of soft masks slightly weakens the effectiveness of the method.
- Limited performance gain. Although GPA only took 200 epochs to achieve similar performance with the state-of-the-art, the performance appears to plateau beyond this point.

**Suitability:**

2

---

### Official Review · Reviewer_TXUA · 2024-05-25

**Rating:** 4
**Confidence:** 3

**Summary:**

This paper focuses on missing high-semantic regions or lacking alignment with the masks and semantics problem of conventional mask strategies, and propose the Global Patch-wise Attention (GPA) framework, a transferable and efficient framework for MIM pre-training. Experiments demonstrate that proposed GPA framework enables MIM to learn better representations, which benefit the model across a wide
range of downstream tasks.

**Strengths:**

This paper designs a highly intuitive and simple method for computing semantic density and utilize it to propose a novel GPA framework, which has a certain degree of innovation. The mask strategy in GPA framework can be applied to any ViT-based masked image modeling approach and consistently improves performance.

**Limitations:**

The effectiveness evaluation of this paper is insufficient. Method validation only compares with classical algorithms, without comparing with the best method on the corresponding dataset, making it difficult to fully demonstrate the effectiveness of this method.
Grammar and formatting need to be more standardized, for example, there is no new paragraph in the "COCO object detection and instance segmentation" section on page 5, line 568.

**Suitability:**

3

---

### Official Review · Reviewer_488j · 2024-05-25

**Rating:** 2
**Confidence:** 4

**Summary:**

The paper designs a GPA framework, including several pivotal steps. Firstly, it defines the Global Patch-wise Attention mechanism, leveraging ViT blocks to calculate attention among individual patches. This enables the identification of regions with high semantic significance within the image. Secondly, the paper describes a Soft-to-Hard Mask Generation Strategy, which involves a gradual transition from soft masks to hard masks. This strategy guides the model's attention towards the identified high semantic regions over time. Lastly, the paper introduces an Additional Task Design called Masked Attention Modeling (MAM). MAM utilizes GPA features for reconstruction purposes, further augmenting the overall performance of the model. These key components collectively contribute to the efficacy and innovation of the GPA framework in masked image modeling.

**Strengths:**

1. The introduction of the GPA framework utilizing global patch attention for identifying high semantic regions is an effective improvement over traditional random or block-wise masking strategies.
2. The implementation of the GPA framework is technically sound, and experimental results validate its consistent improvements across various tasks and datasets.
3. The authors conducted extensive experimental evaluations, including benchmarks such as ImageNet-1K, COCO, and ADE20K, validating GPA's effectiveness across various tasks.

**Limitations:**

1. Although the paper mentions the concept of a teacher, it doesn't provide details on the teacher model used in experiments, nor does it investigate the impact of using different teacher models on performance.

2. While the paper can be understood as a knowledge distillation-based algorithm, the authors do not discuss knowledge distillation in the related works section or reference existing MIM methods based on knowledge distillation[1,2,3].

[1] Masked autoencoders enable efficient knowledge distillers. CVPR 2023

[2] Stare at What You See: Masked Image Modeling without Reconstruction. CVPR 2023

[3] mask again: masked knowledge distillation for masked video modeling, MM 2023

3. In Table.6, the performance improvement brought by GPA is limited;

**Suitability:**

3

---

### Meta-Review · Area_Chair_PUvP · 2024-07-01

**Recommendation:** Accept (Poster)
**Confidence:** 5

**Metareview:**

This paper introduces the Global Patch-wise Attention (GPA) framework for masked image modeling (MIM), aiming to enhance semantic understanding and model performance. GPA leverages ViT blocks to compute attention among patches, facilitating the identification of high-semantic regions within images. Key innovations include a Soft-to-Hard Mask Generation Strategy and Masked Attention Modeling (MAM), which collectively improve model effectiveness and generalization across various tasks and datasets, including ImageNet-1K, COCO, and ADE20K. While the paper demonstrates technical rigor and comprehensive experimentation, concerns were raised about the absence of detailed comparisons with state-of-the-art knowledge distillation (KD) methods in Table 1 or Table 6, despite GPA's alignment with KD principles. Addressing these concerns could further validate the method's efficacy and broaden its applicability beyond MIM. Overall, given the strong technical foundation and addressed concerns, I recommend accepting this submission.